# *Rhodiola rosea* L. Extract, a Known Adaptogen, Evaluated in Experimental Arthritis

**DOI:** 10.3390/molecules28135053

**Published:** 2023-06-28

**Authors:** Frantisek Drafi, Katarina Bauerova, Martin Chrastina, Mohsen Taghdisiesfejír, João Rocha, Rosa Direito, Maria Eduardo Figueira, Bruno Sepodes, Silvester Ponist

**Affiliations:** 1Institute of Experimental Pharmacology and Toxicology, Centre of Experimental Medicine SAS, 841 04 Bratislava, Slovakia; frantisek.drafi@savba.sk (F.D.); martin.chrastina@savba.sk (M.C.); mohsen.taghdisiesfejir@savba.sk (M.T.); exfasipo@savba.sk (S.P.); 2Jessenius Faculty of Medicine in Martin, Comenius University in Bratislava, Malá Hora 10701/4A, 036 01 Martin, Slovakia; 3Faculty of Natural Sciences, Comenius University in Bratislava, Ilkovičova 6, 842 15 Bratislava, Slovakia; 4Faculdade de Farmácia, Universidade de Lisboa, Avenida Professor Gama Pinto, 1649-003 Lisbon, Portugal; jrocha@ff.ulisboa.pt (J.R.); rdireito@ff.ulisboa.pt (R.D.); efigueira@ff.ulisboa.pt (M.E.F.); bsepodes@ff.ulisboa.pt (B.S.); 5Laboratory of Systems Integration Pharmacology, Clinical and Regulatory Science, Research Institute for Medicines of the University of Lisbon (iMED.ULisboa), Avenida Professor Gama Pinto, 1649-003 Lisbon, Portugal

**Keywords:** Rhodiola, arthritis, IL-6, IL-17A, MMP-9, CRP, methotrexate, combination therapy

## Abstract

*Rhodiola rosea L.* extract (RSE) is mostly known for its adaptogen properties, but not for its antiarthritic activities, therefore monotherapy and combination with low-dose methotrexate (MTX) was studied. The collagen-induced arthritis (CIA) model was used to measure the functional score, and the change in hind paw volume (HPV). Both parameters had significant antiarthritic effects. Based on these preliminary results, an adjuvant arthritis (AA) model was further applied to assess another parameters. The experiment included these animal groups: healthy controls, untreated AA, AA administered with RSE (150 mg/kg b.w. daily, *p.o.*), AA administered by MTX (0.3 mg/kg b.w. twice a week, *p.o.*), and AA treated with the combination of RSE+MTX. The combination of RSE+MTX significantly reduced the HPV and increased the body weight. The combination significantly decreased HPV when compared to MTX monotherapy. The plasmatic levels of inflammatory markers (IL-6, IL-17A, MMP-9 and CRP) were significantly decreased by MTX+RSE treatment. The RSE monotherapy didn’t influence any of the inflammatory parameters studied. In CIA, the RSE monotherapy significantly decreased the arthritic parameters studied. In summary, the combination of RSE and sub-therapeutic MTX was significantly effective in AA by improving inflammatory and arthritic parameters.

## 1. Introduction

Rheumatoid arthritis (RA) is a chronic, inflammatory joint disease with a worldwide prevalence of about 5 per 1000 adults. The disease affects women 2 to 3 times more frequently than men and occurs at any age [1,2]. RA is characterised by infiltration of the synovial membrane in multiple joints with T cells, B cells, and monocytes. Molecules, such as receptor activator of nuclear factor κB ligand (RANKL), prostaglandins, and matrix metalloproteinases are induced by pro-inflammatory cytokines, including tumour necrosis factor (TNF) and interleukin (IL)-6. They mediate the signs and symptoms of the disease, including pain and swelling, and degradation of cartilage and bone [3]. Stimulation by RANKL, TNF, and IL-6 generates osteoclasts within the synovial membrane and promotes bony damage [4]. These molecular and cellular events result in clinical disease expression. Progression of joint damage is intrinsically associated with joint swelling [5].

RA pharmacotherapy should closely control inflammation with the goal of low disease activity or inducing remission [6]. In the treatment of RA, methotrexate (MTX) is more effective and less toxic than other already used disease-modifying anti-rheumatic drugs (DMARDs)—auranofin and azathioprine and as effective as but less toxic than sulfasalazine [7,8]. The favourable anti-inflammatory effect of low-dose pulse MTX, given as 7.5–30 mg (approximately 0.3 mg/kg) once weekly orally, subcutaneously, or intramuscularly, was first reported in the 1950s in patients with psoriasis and psoriatic arthritis [9]. The drug has been commonly used in the therapy of recalcitrant psoriasis since the 1960s [9]. The use of MTX in the treatment of RA began during the 1980s [7]. At present, MTX is one of the most frequently used DMARDs, also called slow-acting or symptom-modifying drugs for the treatment of RA [10,11]. Methotrexate has immunosuppressive activity [12], possibly in part as a result of the inhibition of lymphocyte multiplication [13,14,15]. The mechanism(s) of action in the management were not known for a long time. The suggested mechanisms have included immunosuppressive and/or anti-inflammatory effects [12,16]. These latter are mediated, at least in part, by adenosine receptors [17]. Methotrexate is the dihydrofolate reductase (DHFR) inhibitor used most often in a clinical setting as an anticancer drug and as an anti-inflammatory and immunosuppressive agent [18]. As a potent inhibitor of DHFR [19], the rate-limiting enzyme in the production of tetrahydrofolate, MTX decreases the de novo production of purines and pyrimidines and interferes with DNA synthesis [20]. Approximately half of all patients treated with MTX have little or no radiographic progression, although 30% will require additional DMARDs [21].

In recent years, natural drug research for the treatment of RA has made remarkable progress. These natural medicines mainly include flavonoids [22], polyphenols [23], alkaloids [24], glycosides [25], and terpenes [26].

A high number of pharmacological investigations on herbal preparations and isolated constituents from the underground parts of *Rhodiola rosea* L. are published [27,28,29]. Some of the studies (e.g., in models investigating anti-fatigue effects, stress-protective effects, and effects on the nervous system) support the medicinal use of herbal preparations from *Rhodiola rosea* L. as an adaptogen and make the use in this indication plausible [30]. Results from studies performed with isolated constituents suggest that the phenylethanoids like salidroside contribute to these effects [31,32]. Limited data are published on the toxicity of *Rhodiola rosea* L. preparations [29]. An overwhelming accumulation of reactive oxygen species (ROS) leads to various disorders, and it represents distress for living organisms [33,34,35]. At the same time, certain levels of ROS are necessary for physiological redox signalling and essential for proper cellular/tissue/organism functioning [36,37]. Oxidative stress (OS) plays an essential role in the emergence of several chronic disorders, such as diabetes [38] and cancer [39], by inducing inflammation [40]. Moreover, OS is involved in synovial inflammation and contributes to the development and progression of RA [41]. To alleviate OS, antioxidants protect cells against oxidative damage via several mechanisms [42]. Oxidative stress and antioxidative effects of *Rhodiola rosea* L. were studied experimentally by Battistelli et al. (2005). The results obtained from in vitro human erythrocytes exposed to hypochlorous acid (HOCl) are consistent with significant protection of the extract in the presence of the oxidative agent [43]. This was further proven by De Sanctis et al. (2004) in the same in vitro pattern experiment [44]. Keratinocytes of type NCTC 2544 are capable of better counteracting several oxidative insults if incubated with an aqueous extract of *Rhodiola rosea* L. [45]. The anti-inflammatory effects were studied by Pooja et al. (2009), where the anti-inflammatory effects of a liquid extract of underground organs of *Rhodiola rosea* L. (extraction solvent: 40% ethanol) were evaluated. Anti-inflammatory activity was determined in different rat models as carrageenan-induced paw oedema, formaldehyde-induced arthritis, and nystatin-induced paw oedema. The liquid extract exhibited an inhibitory effect against acute and subacute inflammation at a dose of 250 mg/kg body weight. Inhibition of nystatin-induced oedema was also observed in a dose-dependent manner, i.e., 50 mg, 125 mg, and 250 mg/kg body weight of *Rhodiola rosea* L. root was administered intraperitoneally 30 min before inducing oedema. The extract showed varying inhibitory activities against enzymes related to inflammation depending on concentrations. A potent inhibition against COX-2 and phospholipase A2 was observed. Inhibition of nystatin-induced oedema and phospholipase A2 suggested that membrane stabilization could be the most probable mechanism of action of the extract in the acute anti-inflammation effect [46].

From the technological point of view, *Rhodiola rosea* L. extract was also tested for its properties as the influence of the technological processes of RS dry extract on tablets [47]. Haršányová et al. (2020) mixed the dry RS extract with excipients and determined the flow characteristics of a mixture—bulk volume and tapped volume. The authors concluded that in some formulations, the impact of the dry extract’s properties on various technological aspects of the final tablets was more marked [47]. Dry RS extract was also studied for the matrix adhesive system, where the most promising formulations were matrix systems containing gelatine with chitosan and pectin with polyethylene oxide [48].

The purpose and design of the presented study were based on previous experiments, where MTX was used in combination with other natural compounds such as N-feruloyl serotonin [49], ferulaldehyde [50], β-glucan [51], coenzyme Q_10_ [52], pinosylvin and carnosine [53]. The aim of this study was to evaluate the anti-rheumatic and anti-inflammatory effects of a standardized *Rhodiola rosea* L. extract in a model of subacute arthritis. Unlike acute models, it also evaluates the combined therapy with low-dose MTX, which can be described as an original approach. The suitability of using a standardized *Rhodiola rosea* L. dry extract, as a promising extract with anti-rheumatic properties, was studied in a pilot experiment on a collagen-induced arthritis model in rats, a 10-day dosing design. For this pilot experiment, we choose the dose 150 mg/kg based on acute inflammatory experiments with *Rhodiola rosea* L. performed by Pooja et al. (2009) [46]. The CIA experiment was then followed up by a study on the 21-day AA model on Lewis rats with a wider spectrum of inflammatory parameters. The original results of this research from both experiments are presented in this article.

## 2. Results

The first two results are described for the pilot experiment with dry RS extract in the CIA model (Figure 1 and Table 1). The further results of the pivotal experiment performed on the AA model are presented from Figure 2, Figure 3, Figure 4 and Figure 5, including Table 2.


Paw oedema—Plethysmography in CIA


In the preliminary study on the CIA model, we found that the change in hind paw volume (HPV) was significantly increased in the CIA group compared to the healthy control on day 10 after starting the therapy. The administration of RS significantly decreased the percentage of HPV increase on day 10.


Qualitative Functional Scoring System—Severity of Paw Inflammation in CIA


In the preliminary study on the CIA model, we found that the functional score (FS) in CIA was significantly increased in the CIA group compared to the healthy control on day 10. Administration of RS significantly decreased the points of FS on day 10 after starting the therapy—Table 1.


Changes in Hind Paw Volume in AA


The change in hind paw volume (HPV) in untreated AA animals was increased compared to healthy control animals (HC) over time. The increase was significant on days 14 and 21 (AA vs. HC, + *p* > 0.05, Figure 2). This increase indicates that AA was properly induced in this experiment. The change in HPV in methotrexate (MTX) treated animals was significantly decreased, which was observed on days 14 and 21 (AA-MTX vs. AA, * *p* > 0.05, Figure 2). This indicates the effectiveness of the medical standard, MTX. Monotherapy of *Rhodiola rosea* L. extract (RS) was effective in comparison to untreated AA animals on day 14 only. The effect was not observed on day 21 (RS vs. AA, Figure 2). The combination of RS and MTX was significantly effective in comparison to untreated AA animals on days 14 and 21 (RS-MTX vs. AA, * *p* > 0.05, Figure 2). This effect was also significant in comparison to MTX monotherapy (RS-MTX vs. MTX, # *p* > 0.05, Figure 2) on day 21.


Changes of Body Weight in AA


The body weight change in untreated AA animals was significantly decreased on all experimental days compared to the healthy control, (AA vs. HC, + *p* > 0.05, Figure 3), indicating the progression of the disease induced in the untreated AA group. Methotrexate was effective on all days observed compared to untreated AA animals, significantly on days 14 and 21 (AA-MTX vs. AA, * *p* > 0.05, Figure 3). *Rhodiola rosea* L. dry extract monotherapy showed no significant effect in comparison to untreated AA animals. The combination of rhodiola with MTX was slightly more effective than rhodiola monotherapy but less effective than MTX alone (RS-MTX vs. AA, * *p* > 0.05, Figure 3).


Levels of CRP in Plasma Samples on day 21 (AA)


The levels of plasmatic CRP were measured on day 21. The level of CRP in untreated AA animals was significantly elevated compared to the healthy control group (AA vs. HC, + *p* > 0.05, Table 2). Administration with MTX was not effective in influencing the CRP level, similarly as monotherapy with RS. However, the only significant treatment effect was the combination of RS+MTX (RS-MTX vs. AA, * *p* > 0.05, Table 2).


Levels of MMP-9 in Plasma Samples on days 14 and 21 (AA)


Plasma MMP-9 levels in untreated AA animals increased significantly on all experimental days compared to healthy controls (AA vs. HC, + *p* > 0.05, Figure 4). Methotrexate, as well as *Rhodiola rosea* L. dry extract monotherapy, had no effect compared to untreated AA animals. The combination of Rhodiola in combination with MTX was the only treatment that significantly lowered MMP-9 levels on day 14 (RS-MTX vs. AA, * *p* > 0.05, Figure 4).


Levels of Interleukin-17A in Plasma Samples on Days 14 and 21 (AA)


The level of IL-17A was measured on days 14 and 21 in plasma. Untreated AA animals had significantly elevated levels of IL-17A compared to healthy control for both experimental days (AA vs. HC, + *p* > 0.05, Figure 5). Monotherapy of MTX as well as *Rhodiola rosea* L. dry extract was not significantly effective on days 14 and 21 compared to untreated AA animals. However, the combination of MTX and RS significantly decreased the levels of IL-17A on days 14 and 21 (RS-MTX vs. AA, * *p* > 0.05, Figure 5).


Levels of Interleukin-6, in Plasma Samples on days 14 and 21 (AA)


Untreated AA animals had significantly elevated levels of IL-6 in comparison to healthy control for both experimental days (AA vs. HC, + *p* > 0.05, Figure 6). The effect of MTX monotherapy was effective, significantly on day 14 (MTX vs. AA, * *p* > 0.05, Figure 6). *Rhodiola rosea* L. dry extract monotherapy did not have an effect on IL-6 levels. However, the combination of MTX and RS significantly decreased plasma levels of IL-6 levels on both experimental days (RS-MTX vs. AA, * *p* > 0.05, Figure 6).

## 3. Discussion

Rheumatoid arthritis (RA) is an autoimmune and chronic disease that mainly affects joints and causes the destruction of bones [54,55,56,57]. Because RA disease activity is strongly associated with mortality [58,59], survival could have been expected to improve along with changes in treatment paradigms that have occurred in the past few decades, including early intervention [60,61,62,63], and treat-to-target approaches [64]. However, the implementation of these recommendations appears to remain suboptimal. Although many patients are adequately treated with methotrexate (MTX) and other conventional synthetic (cs) disease-modifying anti-rheumatic drugs (DMARDs), an estimated two-fifths of patients do not respond to these first-line treatments and require a biologic (b) or targeted synthetic (ts) DMARD [65]. The EULAR has established a task force on ‘difficult-to-treat RA’ [66], a concept that incorporates uncontrolled inflammatory diseases, but also wider contextual factors such as chronic pain and fatigue, as well as comorbidities, recurrent infections, or, other adverse events that limit treatment options [67]. Based on these recent limitations regarding MTX in clinical praxis, our team decided to investigate the combination with the extract of *Rhodiola rosea* L. in AA. Prior to this pivotal pre-clinical study, we performed a preliminary experiment on collagen-induced arthritis (CIA) model, which will be discussed later. Methotrexate is recently still considered the first-line treatment for RA [67,68] and serves as a medical standard for the evaluation of monotherapy with *Rhodiola rosea* L. dry extract (RSE), as well as combination therapy with its subtherapeutic dose. *Rhodiola rosea* L. has gained attention for its various therapeutic properties which in the EU are used mainly for the indication as an adaptogen [69]. The anti-inflammatory properties of (RSE) in experimental arthritis (CIA and/or AA) have scarcely been studied. Therefore, we decided to investigate the anti-inflammatory and immunomodulatory activity of RSE in monotherapy and in combination with MTX in the AA model to study its ability to enhance MTX treatment. The preliminary experiment was carried out on CIA to demonstrate its antiarthritic ability in a dose of 150 mg/kg of b.w. administered *p.o.* daily for 10 days. Biometric results from the preliminary experiment indicated the beneficial effect of RSE on the change of hind paw volume (Figure 1) and functional score (Table 1), both evaluated on experimental day 10. Administration of RSE showed its efficacy in reducing these parameters significantly compared to untreated CIA animals. These findings are in agreement with previous studies performed by Pooja et al. (2009) and were determined through carrageenan-induced paw oedema, formaldehyde-induced arthritis, and nystatin-induced paw oedema in a rat model [46]. As concluded by the authors of their findings, *R. rosea* root extract might be used in the treatment of acute inflammatory conditions [46].

Based on these promising preliminary results on CIA and literature review, we decided to examine the anti-inflammatory and antiarthritic effects of the same dose of RSE (*p.o.* administration of 150 mg/kg of RSE daily) on AA model in monotherapy as well as in the combination with MTX (*p.o.* administration of 0.3 mg/kg of MTX twice a week) up to 21 days, which could be considered as approaching the chronic phase of the diseases. In the timeframe of weekly measurements (days 14 and 21) two biometric parameters were evaluated in the pivotal experiment on AA, i.e., the change in body weight (Figure 3) and the change in hind paw volume (Figure 2). RSE did not prove its antiarthritic effect for both biometric parameters mentioned above on the AA model in a monotherapeutic setting. RSE monotherapy in the AA model (Figure 2) does not correlate with significant improvement in hind paw volume change in the CIA model (Figure 1). Our team hypothesises that this discrepancy might be due to the different pathophysiological conditions described already in detail by others [70]. The authors used cyclosporine at a dose of 25 mg/kg daily for 14 days (CIA) and adjuvant arthritis in Sprague-Dawley rats during an observation period of 45 days. Results of the experimental study indicate that specific immunologic unresponsiveness can be induced by cyclosporine in the two experimental models of polyarthritis, collagen arthritis, and adjuvant arthritis and that there is no cross-reactivity between type II collagen and the mycobacterial cell wall components. The results further indicate that immunity to type II collagen plays a critical role in the pathogenesis of collagen arthritis but this pathogenetic role in adjuvant arthritis is insignificant [70]. These conclusions were further supported also by other authors [71,72,73], where different pathophysiological features of RA animal models were present. Furthermore, the pathophysiological difference might be explained based on T cell and antibody-mediated autoimmune reactivity against type II collagen, the primary component of cartilage. The CIA model is characterised by rapid and severe cartilage and bone erosion. Suppression of collagen arthritis was achieved with neutralising antibodies against TNFα and with soluble TNF receptors [74,75,76]. Intriguingly, it was found that TNFα was crucial at the beginning of CIA but appeared less dominant in the later stages [77]. In fact, studies in TNF-receptor-knockout mice demonstrated that the incidence and severity of arthritis were lower in such mice; once joints became affected, however, the entire progression to erosive damage was observed in a TNF-independent fashion [78]. IL-1 is a potent cytokine in the induction of cartilage destruction [79,80,81] and a pivotal secondary mediator in arthritis and tissue destruction in TNF transgenic over-expression models [82,83]. Additionally, IL-1 is not necessarily a dominant cytokine in the acute, inflammatory stages of most models of arthritis. IL-1 constantly plays a crucial role in the propagation of joint inflammation and concomitant cartilage and bone erosion in collagen arthritis. Transgenic over-expression of IL-1 produces erosive arthritis [83,84,85]. In our team’s preliminary CIA study, the effect of treatment on the biometric parameter—hind paw volume (Figure 1) might be predominantly mediated by salidroside and other polyphenolic compounds chemically specified in the dry RSE according to the analytical certificate. Consistent with identifying TNFα and IL-1β as separate targets in CIA, we may hypothesise that salidroside and other polyphenolic compounds might affect these two pro-inflammatory cytokines—TNFα and IL-1β. Moreover, it has been convincingly demonstrated that TNFα and IL-1 blockers therapy provides optimal protection against cartilage destruction [86] thus similar to the volume of the hind paw in CIA (Figure 1). The significant decrease in functional score in the CIA model by RSE monotherapy (Table 1) on day 10 (after starting the rats’ treatment) could be explained by the above-mentioned similar hypothesis as well as due to the complex nature of this parameter [87]. Unfortunately, we were unable to find in the literature a study with salidroside in CIA, which could support this theory. Another possible contribution of salidroside in CIA could be due to its anti-inflammatory effects already demonstrated in a model of focal cerebral ischemia in rats [88] and reported cytoprotection including chondrocyte in vitro [89,90].

During the early stages of inflammation in the AA model, the cytokines expressed in the joint include IL-17, IFN, and TNFα, together with cytokines involved in macrophage stimulation [71]. As the severity of inflammation progresses in the joint, increased levels of IL-4, IL-6, monocyte chemotactic protein 1, and TGF-β can be detected and, consequently, TNFα, IL-1β, IL-21, and IL-17 are all involved in the pathology of this disease [91]. Our findings were supported by this observation described above, that is, AA was significantly expressed in untreated AA animals compared to healthy control animals (Table 2; Figure 4, Figure 5 and Figure 6). Focusing on immunological parameters measured on the AA model, such as CRP in plasma on day 21 (Table 2), MMP-9 in plasma on days 14 and 21 (Figure 4), IL-17A (Figure 5) and IL-6 (Figure 6), both in plasma on days 14 and 21, none of these markers of inflammation were affected by dry RSE in dose 150 mg/kg daily. It may be hypothesised that the compounds contained in dry RSE do not influence these parameters on the AA model directly but only via its combination with MTX as we describe later. This monotherapeutic ineffectiveness of RSE in the AA model could also be explained by cortisol release blockade. Cortisol and consequently corticoids are known for their immunosuppressive activity in RA [92]. Rhodiola has been described to be able to blunt cortisol release [93] and thus influence the anti-inflammatory effect. However, a low to the normal-low baseline level of cortisol was found to be a good predictor of possible MTX efficacy in the treatment of rheumatoid arthritis. On the other hand, high baseline cortisol levels are a predictor of a failure of drug efficacy. As a result of effective MTX treatment, it was found that the ratio of serum cortisol to adrenocorticotropic hormone correlated inversely with patient improvement and, in general, cortisol levels decrease throughout effective treatment [94]. This finding could explain the better efficacy of combination treatment and the ineffective monotherapy with RSE.

In addition, both biometric parameters on the AA model were affected by the combination of dry RSE with MTX. CBW (Figure 3) was significantly elevated on days 14 and 21 compared to untreated AA animals. Surprisingly, HPV (Figure 2) decreased significantly on day 21 compared to untreated AA, as well as compared to AA animals treated with MTX monotherapy. To the best of our knowledge, this was the first time we have demonstrated the beneficial and significant contribution of the combination of dry RSE with MTX in this key biometric marker. The active component of dry RSE, salidroside, has various biological properties, mediating antioxidation [95,96] and lowering inflammation [97]. Salidroside was identified in our RSE batch No. 3-049-001-06-18 by an independent analysis at a commercial analytical laboratory—Calendula, a.s. (see Section 5.5 Phytochemical composition of the *Rhodiola rosea* L. dry extract (RSE)) and as an established component also by other authors [98,99]. Together with the complex of other polyphenols in dry RSE, this could modulate the MTX add-on combination to achieve a significant decrease in HPV on day 21 in the AA model (Figure 2). This complement effect of natural compounds to the antiarthritic contribution of MTX is supported by findings of other authors [100,101] and is in line with our team’s previous experiments [49,50,53,102].

The significant effect of the combination of dry RSE and MTX was observed during days 14 and 21 on interleukin 6 (Figure 6) and interleukin 17A (Figure 5). However, these effects were not significant compared to MTX monotherapy. Thus, it may be hypothesised that the slight effect of this dry RSE contribution might be explained by its salidroside and polyphenolic compounds. The study by Guo et al. (2022) describes that IL-17A is associated with immune cells infiltration in hyperoxia-induced acute lung injury (HALI) and contributes to ferroptosis of type II alveolar epithelial cells that is related to Act1/TRAF6/p38 MAPK pathway. The authors concluded that the salidroside protects against HALI throughout the entire pathogenic process of alveolar epithelial cells and thus modulates the IL-17A indirectly [103]. Furthermore, IL-17A plays an important role in different forms of cell death, including apoptosis [104]. In the presence of interleukin (IL)-6 together with TGF-β, naïve CD4 + T cells may differentiate into Th17 cells. As an important pro-inflammatory factor, IL-17A is secreted mainly by Th17 cells and is closely related to many diseases, including autoimmune diseases [105]. The effect of the combination on plasmatic levels of both interleukins on day 21 could be related as previously described and significantly affected by the compounds present in dry RSE in AA. The long-term administration of dry RSE might cause a more profound effect of the combination on day 21, which has already been described in humans for other indications, such as fatigue, weakness, exhaustion, irritability, and slight anxiety. The onset of the *Rhodiola rosea* L. extract was clinically described from days to weeks after oral administration. While some researchers have found that *Rhodiola rosea* L. could take effect in a matter of days, most studies [106] and clinical trials [107] showed that it takes about four weeks to observe improvements.

Elevated MMP-9 levels can be found in cases of rheumatoid arthritis [108]. To study the effect of dry RSE in monotherapy and in combination with MTX, we also measured the plasmatic level of MMP-9 in this experiment. The decrease in MMP-9 level was detected by the combination on day 21 and significantly on day 14 (Figure 4). This finding is in agreement with the ability of RSE to reduce the expression of matrix metalloproteinase-9 (MMP-9) in a dose-dependent manner [109] and with the ability of salidroside to inhibit the MMP-9 and MMP-2 in HT1080 cells [110]. A similar observation was found with saffron extract in combined therapy with MTX and the combination of carnosic acid with MTX in AA, both combinations have significantly decreased plasmatic levels of MMP-9 [111].

On the other hand, the level of CRP was decreased significantly by the combination only on day 21 in comparison to untreated AA animals (Table 2). The so-called acute phase response mediated by CRP occurs due to increasing concentrations of IL-6, produced by macrophages [112] as well as adipocytes [113] in response to a wide range of acute and chronic inflammatory conditions such as infections, rheumatic and other inflammatory diseases [114]. CRP also appears to influence osteoclast activity leading to bone resorption and stimulating RANKL expression in peripheral blood monocytes [115]. In a mice diabetes model, treatment with *Rhodiola rosea* L. decreased the LPS levels in the serum [116]. Although this result is inconsistent with our team’s findings regarding RSE monotherapy, it shows that in a diabetic model with less intense inflammation than in the AA model, RSE can reduce CRP levels. This is probably why the combination with MTX effectively decreased the CRP level, but not RSE monotherapy. Considering the decrease in IL-6 (Figure 6), the marker correlate with the above-mentioned dependence on day 21, and it may be hypothesised that this effect is due to the additional contribution of dry RSE in combination with MTX. Researchers have found that in LPS-induced inflammation in murine macrophages J774.1 (salidroside in doses 5, 25, 125 µg/mL) [117] and in RAW264.7 (salidroside in doses 50, 100, 200 μM) [118] pre-treatment with salidroside could decrease the level of IL-6, IL-1β, TNFα, NO and MCP-1 through NF-κB signalling pathway and may be a key component of the additional contribution of RSE to MTX.

To explain the effect of RSE in combination with MTX in the AA model it could be hypothesised that this effect might be mediated by RSE inhibition of P-glycoprotein (P-gp). Hellum et al. (2010) investigated *Rhodiola rosea* L. for its in vitro inhibitory potential on CYP3A4-mediated metabolism and P-gp efflux transport activity. *Rhodiola rosea* L. showed potent inhibition of CYP3A4 and P-gp activities, with IC50 values ranging from 1.7 to 3.1 μg/mL and from 16.7 to 51.7 μg/mL, respectively [119]. From the MTX metabolism, it is known that the intracellular conversion of methotrexate to polyglutamate represents the most important metabolic pathway regarding efficacy [9,120,121]. Polyglutamyl derivatives of methotrexate also exhibit more efficient inhibitory properties toward intracellular metabolism of pyrimidines and purines than do the parent drug—MTX [9,121,122]. The role of p-glycoprotein in resistance to MTX and its efficacy plays a vital role, as it has been described experimentally in rats [123] and also in vitro [124]. To support our team’s hypothesis, it has been also shown that naturally occurring flavonoid nobiletin reverses MTX resistance via inhibition of P-glycoprotein synthesis in CIA-FLS/MTX cells through the PI3K/AKT/HIF-1α pathway [125].

## 4. Conclusions

The *Rhodiola rosea* L. dry extract has shown its antiarthritic and anti-inflammatory effect in two models of experimental arthritis. The preliminary study on the CIA model has shown that the monotherapy in a daily dose of 150 mg/kg is effective on biometric parameters—the paw oedema as well as the functional score—as the primary antiarthritic effect. Conversely, the pivotal experiment on the AA model did not confirm that *Rhodiola rosea* L. dry extract monotherapy is able to modify biometric and/or immunological parameters. Interestingly, the combination with a sub-therapeutic dose of methotrexate (0.3 mg/kg twice a week) was more effective than methotrexate monotherapy in the change of the hind paw volume, demonstrating that with reduced doses of methotrexate (and consequently less side effects) a better response may be obtained when in combination with *Rhodiola rosea* L. All other parameters on the AA model were significantly modified by this combination in comparison to the untreated AA group. These various effects might be explained by different experimental conditions in the pathophysiology of both models as well as the salidroside ability to control adjuvant arthritis together with the possible pharmacokinetic interaction of the active substances in the dry extract and methotrexate on P-glycoprotein and inhibition of CYP3A4 and further also by controlling the inflammation mediated by cortisol. The authors would like to study the predominant components of RSE, responsible for those as mentioned for above in vivo effects, by theoretical methods (in particular, DFT) as described by Demirpolat et al. (2023) as well [126]. Finally we supposed that preclinical studies will be followed by clinical evaluation according to current directives.

## 5. Materials and Methods

### 5.1. Animals and Experimental Model of Adjuvant Arthritis

In this experiment, Lewis male rats were purchased from the Department of Toxicology and Laboratory Animal Breeding, Centre of Experimental Medicine, SAS, Dobrá Voda, Slovak Republic (SK CH 24016). Immediately after the housing of animals, rats were placed in a seven-day quarantine. Animals had unlimited access to a standard diet and tap water *ad libitum*, as well as a dark/light regime of 12 h/12 h. Animal housing was in accordance with the EU Convention for the Protection of Vertebrate Animals Used for Experimental and Other Purposes. The authorization of the protocol for this experiment was obtained from the Ethics Committee of the Institute of Experimental Pharmacology and Toxicology, Center of Experimental Medicine SAS in Bratislava, Slovakia (SK UCH 04018) and the State Veterinary and Food Administration of the Slovak Republic, Bratislava (3144/16-221/3). In accordance with Directive 2010/63/EU [127], we have implemented the principle of the 3Rs (Replacement, Reduction, and Refinement). The application of the 3Rs is currently also embedded in scientific guidance at the European and international ICH (International Cooperation on Harmonisation of Technical Requirements) levels [128]. In our in vivo experiments, we have also followed these current scientific principles.

### 5.2. Adjuvant Arthritis (AA) in Lewis Rats

AA is a well-established model of inflammation [71,76]. It is suitable for studying molecular mechanisms between T cells and subpopulations [129]. Then again, this model lacks the chronic progressive nature of RA, and the pathological changes are self-limiting [130]. Adjuvant arthritis was induced in rats with a weight of 160–180 g (6 weeks) by individual intradermal immunization at the base of the tail with a suspension of 0.1 mL of 12 mg/mL heat-inactivated *Mycobacterium butyricum* powder suspended in incomplete Freund’s adjuvant according to our previous protocol [131,132].

### 5.3. The Design of the AA Experiment

The rats were randomised into five experimental groups. Group one was used as healthy control. The second was the untreated AA group. The remaining three groups were AA rats treated as given in the study design below:HC—(healthy control)—vehiculumAA—(animals with induced AA)—vehiculumAA-MTX—methotrexate 0.3 mg/kg twice a weekRS—*Rhodiola rosea* L. dry extract 150 mg/kg dailyRS-MTX—*Rhodiola rosea* L. dry extract 150 mg/kg daily and MTX 0.3 mg/kg twice a week.

The tested substance and MTX were administered orally (via gastric tube) throughout the entire experiment (21 days); *Rhodiola rosea* L. dry extract (RSE) was administered orally throughout the experiment (21 days); anti-rheumatic drug—MTX was diluted with tap water and applied two times a week in a dose of 0.3 mg/kg of b.w. On days 14 and 21, blood was drawn into heparinised tubes from the rat’s retro-orbital plexus using tiletamine, zolazepam plus, and xylazine anaesthesia. On the last experimental day, rats were sacrificed in anaesthesia and blood was collected from all animals. Blood samples were centrifuged for 15 min to obtain plasma. The samples were stored at −80 °C.

### 5.4. Evaluation of Experimental AA

On the 14th and 21st days after immunization of the animals, the volume of hind paw joints was assessed. Hind paw volume (HPV) was expressed as the average elevation of percentage (%) of hind paw volume of each rat, compared to HPV measured on day 1 using a water plethysmometer (UGO BASILE, Comerio-Varese, Italy). The HPV on the selected day was divided by the HPV on day 1 and expressed in a percentage according to the following formula: ([Day n]/[Day 1]) × 100 − 100 = value [%].

The body weight of the animals was measured daily with the aim to precise the dosing. The changes in body weight on day 14 and 21 were calculated as follows: [Day n] − [Day 1] = value [g].

### 5.5. Phytochemical Composition of the Rhodiola rosea L. Dry Extract (RSE)

The origin of the substance applied was the dry extract of RSE from the root, extraction solvent was ethanol 60% m/m. This *Rhodiola rosea* L. dry extract was manufactured by Calendula, Slovakia, and gifted as a courtesy to the Faculty of Pharmacy at Comenius University in Bratislava. Flowingly, the *Rhodiola rosea* L. dry extract, batch No. 3-049-001-06-18, was than gifted to our department, where one part of this batch was tested at Faculdade de Farmácia da Universidade de Lisboa, in Lisbon, Portugal for testing on the collagen-induced arthritis and the second part of the same batch was tested at the Centre of Experimental Medicine SAS, Institute of Experimental Pharmacology and Toxicology, v.v.i, Bratislava, Slovakia for experiments on the adjuvant-induced arthritis. Batch No. 3-049-001-06-18 was independently analysed at a commercial analytical laboratory—Calendula, a.s., Nová Ľubovňa 238/A, 065 11. The analytical conditions were in accordance with the Ph.Eur.5.1.4, where the parameters analysed met the Ph.Eur. criteria. Next, the chemical properties were analysed, from which the dry matter content was 97 ± 1% (from min. 90% of the required value). Additionally, the content of salidroside (rhodioloside) was 2.6 ± 0.2% (by HPLC) and the value of other polyphenolic compounds analysed was 45 ± 2% (spectrophotometrically; from min. 30% of the required value). The physical properties were:Water soluble → solubleParticle size → 0.25 mm (60 mesh)Bulk density → 0.56 g/mL (from the range of 0.20–0.60 g/mL of the required value)Ash → 3.5 ± 0.5% (from max. 15% of the required value)

No preservatives or antioxidants were found in the dry extract. The excipient in batch No. 3-049-001-06-18 was 20% maltodextrin. The identity test of the dry RSE was confirmed by thin-layer chromatography (Ph. Eur. 7) for separations and identification of the minimum content of a particular herbal drug content.

### 5.6. Immunological Evaluation of AA

Immunological changes in plasma from AA animals were evaluated on the 14th and 21st day of the experiment for IL-17A, MMP-9, and for IL-6. The CRP marker was measured on the 21st experimental day. On the 14th and 21st day, when the experiment was terminated, heparinised blood was collected for further plasma evaluation: levels of CRP were determined by a commercial ELISA kit (Immunology consultant laboratories, Inc., Portland, OR, USA) as well as plasmatic levels of IL-17A, MMP-9, and IL-6. All plasma samples were stored at −80 °C until immunological ELISA analysis.

### 5.7. Collagen-Induced Arthritis (CIA) Induction

CIA was once recognised as the best model for RA, but only under certain experimental conditions. On the other hand, the CIA model cannot show RA fluctuations and recurrences, vasculitis symptoms, subcutaneous nodules, or serositis [130]. The CIA study was carried out using 18 male Wistar rats weighing 200 to 250 g (IHMT, Lisbon, Portugal). All rats received a standard diet and water ad libitum. Bovine CII was dissolved in 0.01 M acetic acid at a concentration of 2 mg/mL by stirring overnight at 4 °C. The dissolved CII was frozen at −70 °C until it was used. The complete Freund’s adjuvant (FCA) was prepared by adding *Mycobacterium tuberculosis* H37Ra at a concentration of 2 mg/mL. Before injection, CII was emulsified with an equal volume of FCA.

### 5.8. Design of Collagen-Induced Arthritis

Animals were randomly assigned to 3 groups: 1. sham group (n = 4), 2. Untreated collagen-induced arthritis (CIA) (n = 7), Collagen-induced arthritis administered with *Rhodiola rosea* L. dry extract (150 mg/kg/day) (n = 7). CIA was induced as previously described by Figueira et al. (2014) [133]. On day 1, all rats were administered intradermally at the base of the tail with 100 μL of the emulsion, containing 100 μg of type II collagen (except for the sham group which was administered intradermally with a 100 μL saline solution). On day 21, the second injection of CII in FCA was administered. In treated groups, animals were orally administrated with the *Rhodiola rosea* L. dry extract (150 mg/kg/day) by gastric gavage every 24 h, starting on day 25. The rats in the Sham and CIA groups were administered orally with a vehicle. Rats were evaluated regularly for weight, external signs of distress, and arthritis evolution. Functional severity was also determined by quantifying the change in paw volume, measured by plethysmography on day 35. A Qualitative scoring system was also used to assess the severity of paw inflammation and its impact in paw/mobility function.

### 5.9. Qualitative Functional Scoring System in CIA

The system of qualitative scoring, which was used to assess the severity of CIA was established by Bendele et al. (1999) [87] and described in detail below. The severity of paw inflammation is summarised in Table 3. For each animal, the paw inflammation score was determined (0 to 4) and summed (Table 3).

The description of the scoring of pannus, cartilage damage, and bone resorption with the points is described as follows:

Pannus: 0 (normal); 1 (minimal infiltration of pannus in cartilage and subchondral bone); 2 (mild infiltration of a marginal zone with minor cortical and medullary bone destruction); 3 (moderate infiltration with moderate hard tissue destruction); 4 (marked infiltration with marked destruction of joints); 5 (severe infiltration with total or near total destruction of joints).

Cartilage damage: 0 (normal); 1 (minimal to mild loss of toluidine blue staining); 2 (mild loss of toluidine blue staining with superficial chondrocyte loss); 3 (moderate loss of toluidine blue staining with multifocal moderate chondrocyte loss); 4 (marked loss of toluidine blue staining with superficial chondrocyte loss with marked chondrocyte loss); 5 (severe and diffuse loss of toluidine blue staining with superficial chondrocyte loss)

Bone resorption: 0 (normal); 1 (small areas of periosteal and marginal zone resorption); 2 (more small areas of periosteal and marginal zone resorption), 3 (obvious resorption of medullary trabeculae and cortical bone); 4 (full thickness defects in the cortical bone and marked loss of medullary bone); 5 (full thickness defects in the cortical bone and destruction of joint architecture).

For each animal, all the above-mentioned scores were determined (0 to 5) and then summed [87].


Statistical Analysis for CIA and AA experiment


The data were expressed as arithmetic mean ± SEM, with n = 7–8 animals (for AA) and n = 4–7 animals (for CIA) in a particular group per experiment. The AA group was compared with HC (*), and AA-MTX, RS1, and RS-MTX groups were compared with AA (+). The CIA group was compared with HC (*) and the RS group was compared with CIA (+). For significance calculations, Welch’s ANOVA (two samples, unequal variance) was used with the following significance designations: significant (*p* < 0.05); not significant (*p* > 0.05).

## Figures and Tables

**Figure 1 molecules-28-05053-f001:**
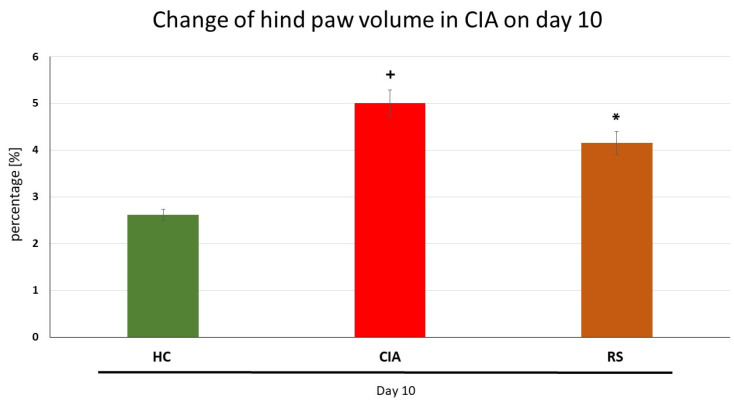
The effect of *Rhodiola rosea* L. extract administered alone on hind paw volume (HPV) on day 10. HC—control animals, CIA—collagen-induced arthritis untreated control, RS— collagen-induced arthritis animals treated with *Rhodiola rosea* L. extract 150 mg/kg daily. Results were expressed as mean ± SEM, n = 4–7. Significant difference: + *p* < 0.05 CIA vs. HC and * *p* < 0.05 treated group vs. CIA.

**Figure 2 molecules-28-05053-f002:**
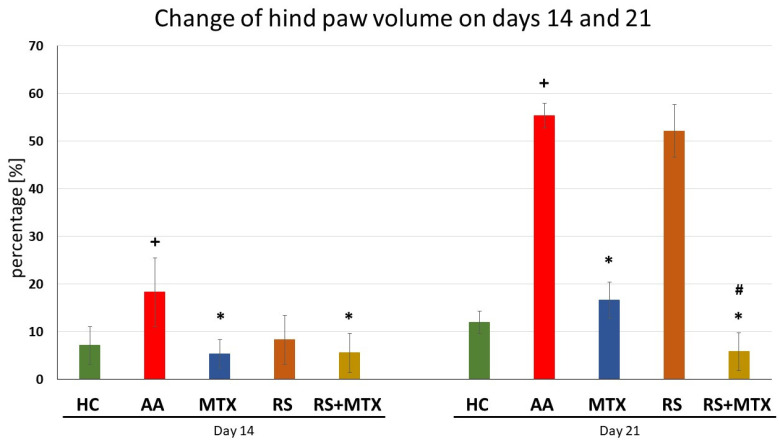
The effect of *Rhodiola rosea* L. extract administered alone or in combination with MTX on hind paw volume (HPV) on days 14 and 21. HC—control animals, AA—adjuvant arthritis untreated control, MTX—adjuvant arthritis group treated with methotrexate 0.3 mg/kg of b.w. twice a week, RS—adjuvant arthritic animals treated by *Rhodiola rosea* L. extract 150 mg/kg daily, RS-MTX—adjuvant arthritic animals treated with RS (150 mg/kg) daily and 0.3 mg/kg of b.w. twice a week. The results were expressed as mean ± SEM, n = 6–8. Significant difference: + *p* < 0.05 AA vs. HC, * *p* < 0.05 treated group vs. AA, and # *p* > 0.05 RS-MTX vs. MTX.

**Figure 3 molecules-28-05053-f003:**
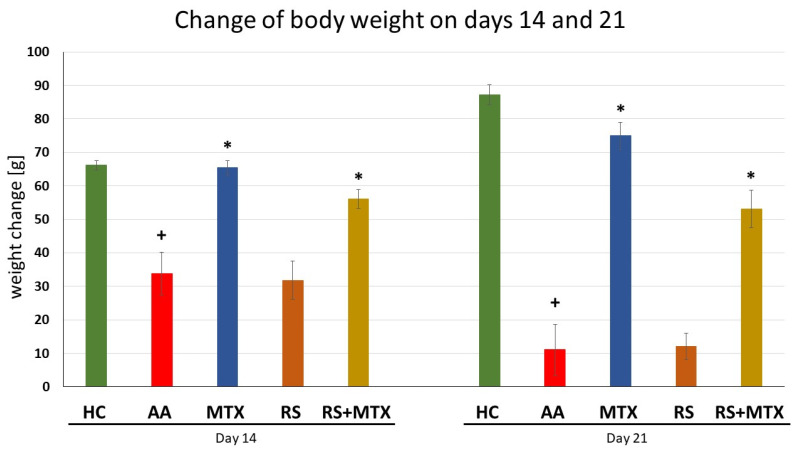
The effect of *Rhodiola rosea* L. extract administered alone or in combination with MTX on change of body weight measured on days 14, and 21. HC—control animals, AA—adjuvant arthritis controls, AA-MTX—adjuvant arthritis group treated with methotrexate 0.3 mg/kg of b.w. twice a week, AA-RS—adjuvant arthritic animals treated with *Rhodiola rosea* L. extract 150 mg/kg daily, RS-MTX—adjuvant arthritic animals treated with RS (150 mg/kg) daily and 0.3 mg/kg of b.w. twice a week. The results were expressed as mean ± SEM, n = 6–8. Significant difference: + *p* < 0.05 AA vs. HC, * *p* < 0.05 treated group vs. AA.

**Figure 4 molecules-28-05053-f004:**
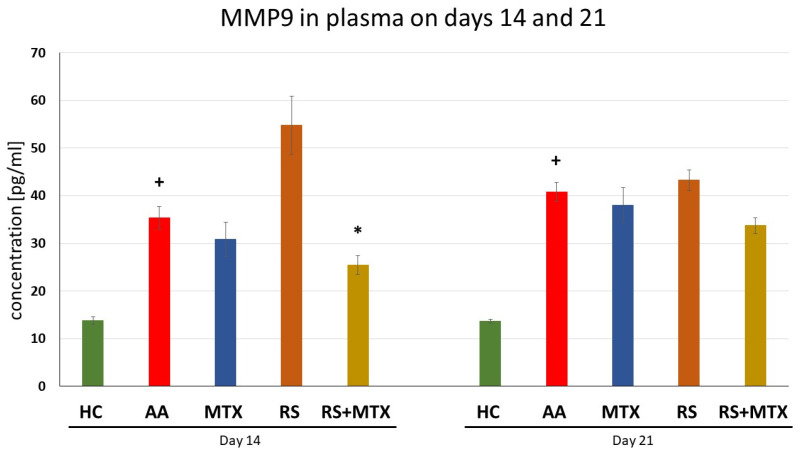
The effect of *Rhodiola rosea* L. extract administered alone or in combination with MTX on the level of MMP9 in plasma was measured on days 14 and 21. HC—control animals, AA—adjuvant arthritis controls, AA-MTX—adjuvant arthritis group treated with methotrexate 0.3 mg/kg of b.w. twice a week, AA-RS—adjuvant arthritic animals treated with *Rhodiola rosea* L. extract 150 mg/kg daily, RS-MTX—adjuvant arthritic animals treated with RS (150 mg/kg) daily and 0.3 mg/kg of b.w. twice a week. Results were expressed as mean ± SEM, n = 6–7. Significant difference: + *p* < 0.05 AA vs. HC, * *p* < 0.05 treated group vs. AA.

**Figure 5 molecules-28-05053-f005:**
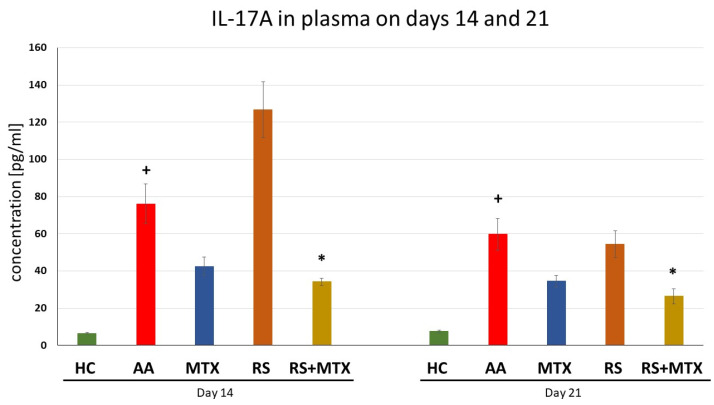
The effect of *Rhodiola rosea* L. extract administered alone or in combination with MTX on the level of IL-17A in plasma was measured on days 14 and 21. HC—control animals, AA—adjuvant arthritis controls, AA-MTX—adjuvant arthritis group treated with methotrexate 0.3 mg/kg of b.w. twice a week, AA-RS—adjuvant arthritic animals treated with *Rhodiola rosea* L. extract 150 mg/kg daily, RS-MTX—adjuvant arthritic animals treated with RS (150 mg/kg) daily and 0.3 mg/kg of b.w. twice a week. Results were expressed as mean ± SEM, n = 6–7. Significant difference: + *p* < 0.05 AA vs. HC * *p* < 0.05 treated group vs. AA.

**Figure 6 molecules-28-05053-f006:**
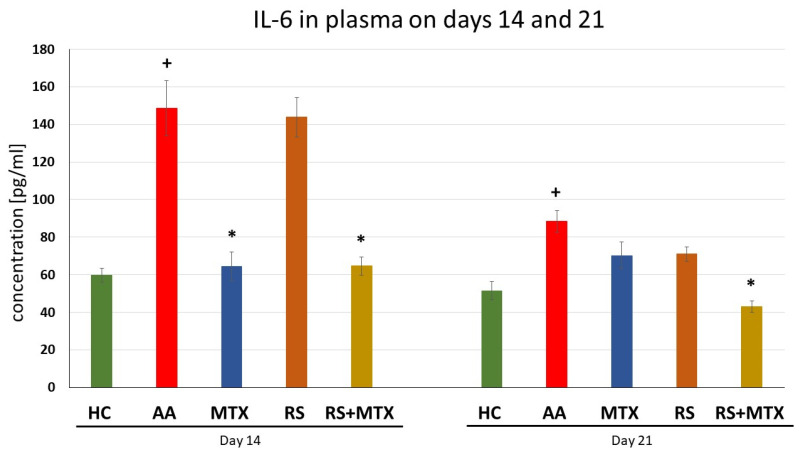
The effect of *Rhodiola rosea* L. extract administered alone or in combination with MTX on the level of IL-6 in plasma was measured on days 14 and 21. HC—control animals, AA—adjuvant arthritis controls, AA-MTX—adjuvant arthritis group treated with methotrexate 0.3 mg/kg of b.w. twice a week, AA-RS—adjuvant arthritic animals treated with *Rhodiola rosea* L. dry extract 150 mg/kg daily, RS-MTX—adjuvant arthritic animals treated with RS (150 mg/kg) daily and 0.3 mg/kg of b.w. twice a week. Results were expressed as mean ± SEM, n = 6–7. Significant difference: + *p* < 0.05 AA vs. HC * *p* < 0.05 treated group vs. AA.

**Table 1 molecules-28-05053-t001:** The effect of *Rhodiola rosea* L. extract administered alone on the functional score on day 10.

Experimental Group	Arithmetic Mean ^1^	SEM ^2^
HC	0.00	± 0.00
CIA	3.71 ^+^	± 0.18
RS	2.29 *	± 0.18

HC—control animals, CIA—untreated collagen-induced arthritis, RS—collagen-induced arthritis animals administered with *Rhodiola rosea* L. extract 150 mg/kg daily. ^1^ n = 4–7, ^2^ Standard Error of the Mean, ^+^ *p* < 0.05 CIA vs. HC, * *p* < 0.05 treated group vs. CIA.

**Table 2 molecules-28-05053-t002:** The effect of *Rhodiola rosea* L. extract administered alone or in combination with MTX on the level of plasmatic CRP was measured on day 21.

Experimental Group	Concentration Arithmetic Mean ^1^ [µg/mL]	SEM ^2^
HC	762.1	±76.9
AA	1775.1 **^+^**	±141.0
MTX	1913.0	±80.2
RS	2544.8	±269.9
RS-MTX	929.2 *	±109.3

HC—control animals, AA—adjuvant arthritis controls, AA-MTX—adjuvant arthritis group treated with methotrexate 0.3 mg/kg of b.w. twice a week, AA-RS—adjuvant arthritic animals treated with *Rhodiola rosea* L. extract 150 mg/kg daily, RS-MTX—adjuvant arthritic animals treated with RS (150 mg/kg) daily and 0.3 mg/kg of b.w. twice a week. ^1^ n = 6–7, ^2^ Standard Error of the Mean, ^+^ *p* < 0.05 AA vs. HC, * *p* < 0.05 treated group vs. AA.

**Table 3 molecules-28-05053-t003:** The below qualitative scoring system was used to assess the severity of paw inflammation.

Score	Condition
0	Normal
1	Mild, but definite redness and swelling of the ankle or wrist, or apparent redness and swelling limited to individual digits, regardless of the number of affected digits
2	Moderate redness and swelling of ankle or wrist
3	Severe redness and swelling of the entire paw including digits
4	Maximally inflamed limb with involvement of multiple joints (mobility severely impaired)

## Data Availability

The following supporting information can be downloaded at: https://figshare.com/articles/dataset/Rhodiola_source_data/22128581. The dataset was posted on 21 February 2023, and authored by Martin Chrastina.

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
