# Peer review of "Rhodiola rosea L. Extract, a Known Adaptogen, Evaluated in Experimental Arthritis"

_molecules, 2023, doi:10.3390/molecules28135053_

Round 1
Reviewer 1 Report
The manuscript entitled as 'Rhodiola rosea L. extract, a known adaptogen, evaluated in experimental arthritis' is well designed and written. However to improve it, suggestions and/or corrections were indicated in the manuscript pdf file.

Author Response
Reviewer 1:
Before mentioning the antioxidative effects of Rhodiola roses. It should be given some details on oxidative stress. In this part of the text oxidative stress and reactive oxygen species should be detailed. Overwhelming accumulation of reactive oxygen species (ROS) leading to diverse disorders represents distress, while levels of ROS necessary for physiological redox signaling important for proper cellular/tissue/organism functioning. 1-Oxidative stress plays an essential role in the emergence of a number of chronic disorders such as diabetes and cancer by inducing inflammation.
According to the Acaroz, U., Ince, S., Arslan-Acaroz, D., Gurler, Z., Demirel, H. H., Kucukkurt, I., ... & Zhu, K. (2019). Bisphenol-A induced oxidative stress, inflammatory gene expression, and metabolic and histopathological changes in male Wistar albino rats: protective role of boron. Toxicology research, 8(2), 262-269.
2-Jaganjac, M., Milkovic, L., Zarkovic, N., & Zarkovic, K. (2022). Oxidative stress and regeneration. Free Radical Biology and Medicine, 181, 154-165.
This references should be added.
Authors’ response:
The authors would like to thank to the Reviewer 1 and the suggestion to add some more detailed information about the OS. Even the fact, we do not focused on the OS and any OS markers in our experimental part, we have modified and added up-to-date text together with suggested citations mentioned by the Reviewer 1 and for contextualization, more references have been implemented:
An overwhelming accumulation of reactive oxygen species (ROS) leads to various disorders, and it represents distress for living organisms (Acaroz U, Ince S, Arslan-Acaroz D, Gurler Z, Demirel HH, Kucukkurt I, Eryavuz A, Kara R, Varol N, Zhu K. Bisphenol-A induced oxidative stress, inflammatory gene expression, and metabolic and histopathological changes in male Wistar albino rats: protective role of boron. Toxicol Res (Camb). 2019 Jan 23;8(2):262-269.9; Ngo V, Duennwald ML. Nrf2 and Oxidative Stress: A General Overview of Mechanisms and Implications in Human Disease. Antioxidants (Basel). 2022 Nov 27;11(12):2345.; Chaudhary P, Janmeda P, Docea AO, Yeskaliyeva B, Abdull Razis AF, Modu B, Calina D, Sharifi-Rad J. Oxidative stress, free radicals and antioxidants: potential crosstalk in the pathophysiology of human diseases. Front Chem. 2023 May 10;11:1158198.). At the same time, certain levels of ROS are necessary for physiological redox signalling and essential for proper cellular/tissue/organism functioning (Jaganjac M, Milkovic L, Zarkovic N, Zarkovic K. Oxidative stress and regeneration. Free Radic Biol Med. 2022 Mar;181:154-165.; Zhang B, Pan C, Feng C, Yan C, Yu Y, Chen Z, Guo C, Wang X. Role of mitochondrial reactive oxygen species in homeostasis regulation. Redox Rep. 2022 Dec;27(1):45-52.). Oxidative stress plays an essential role in the emergence of several chronic disorders, such as diabetes (Ramos-Riera KP, Pérez-Severiano F, López-Meraz ML. Oxidative stress: a common imbalance in diabetes and epilepsy. Metab Brain Dis. 2023 Mar;38(3):767-782.) and cancer (Acevedo-León D, Monzó-Beltrán L, Pérez-Sánchez L, Naranjo-Morillo E, Gómez-Abril SÁ, Estañ-Capell N, Bañuls C, Sáez G. Oxidative Stress and DNA Damage Markers in Colorectal Cancer. Int J Mol Sci. 2022 Oct 1;23(19):11664.), by inducing inflammation (Zhang Z, Zhao L, Zhou X, Meng X, Zhou X. Role of inflammation, immunity, and oxidative stress in hypertension: New insights and potential therapeutic targets. Front Immunol. 2023 Jan 10;13:1098725. ). Moreover, OS is involved in synovial inflammation and contributes to the development and progression of rheumatoid arthritis (Zamudio-Cuevas Y, Martínez-Flores K, Martínez-Nava GA, Clavijo-Cornejo D, Fernández-Torres J, Sánchez-Sánchez R. Rheumatoid Arthritis and Oxidative Stress. Cell Mol Biol (Noisy-le-grand). 2022 Jun 30;68(6):174-184.).
-----------------------------------------------------------------------
Reviewer 1:
It should be also added that: antioxidants protect cells against oxidative damage. The novel refernces should be added:
Ramazani, N., Mahd Gharebagh, F., Soleimanzadeh, A., Arslan, H. O., Keles, E., Gradinarska‐Yanakieva, D. G., ... & Dinç, D. A. (2023). The influence of L‐proline and fulvic acid on oxidative stress and semen quality of buffalo bull semen following cryopreservation. Veterinary Medicine and Science.
Authors’ response:
The authors have implemented the novel reference in the context of a new sentence.
To alleviate OS, antioxidants protect cells against oxidative damage via several mechanisms (Ramazani N, Mahd Gharebagh F, Soleimanzadeh A, Arslan HO, Keles E, Gradinarska-Yanakieva DG, Arslan-Acaröz D, Zhandi M, Baran A, Ayen E, Dinç DA. The influence of L-proline and fulvic acid on oxidative stress and semen quality of buffalo bull semen following cryopreservation. Vet Med Sci. 2023 May 17. doi: 10.1002/vms3.1158. Epub ahead of print.).
----------------------------------------------------------------------------------------
Reviewer 1:
This part of the text should be supported with novel and accurate references.
Authors’ response:
The authors appreciate the suggestion of the Reviewer 1 and would like to implement these novel and accurate references into the particular part, the Reviewer marked:
- Senthelal S, Li J, Ardeshirzadeh S, Thomas MA. Arthritis. 2022Jun 19. In: StatPearls [Internet]. Treasure Island(FL): StatPearls Publishing; 2023 Jan–. PMID: 30085534.
- Chauhan K, Jandu JS, Brent LH, Al-Dhahir MA. Rheumatoid Arthritis. 2023May 25. In: StatPearls [Internet]. Treasure Island (FL): StatPearls Publishing; 2023 Jan–. PMID: 28723028.
- Black RJ, Lester S, Tieu J, Sinnathurai P, Barrett C, Buchbinder R, Lassere M, March L, Proudman SM, Hill CL. Mortality Estimates and Excess Mortality in Rheumatoid Arthritis. Rheumatology (Oxford). 2023 Mar 15:kead106. doi: 10.1093/rheumatology/kead106. Epub ahead of print. PMID: 36919770.
- Kaban N, Harman H. Paradigm guiding to tapering or discontinuation of biologic and targeted synthetic disease-modifying antirheumatic drugs in the treatment of patients with rheumatoid arthritis: Results from a local prospective study. Int J Rheum Dis. 2023 Apr;26(4):689-698.
- Novella-Navarro M, Ruiz-Esquide V, Torres-Ortiz G, Chacur CA, Tornero C, Fernández-Fernández E, Monjo I,Sanmartí R, Plasencia-Rodríguez C, Balsa A. A paradigm of difficult-to-treat rheumatoid arthritis: subtypes and early identification. Clin Exp Rheumatol. 2023 May;41(5):1114-1119.
-----------------------------------------------------------------------------------------
We hope all issues were addressed and met for this manuscript. We believe that the manuscript has now been sufficiently improved, and therefore, the authors would also like to express their thanks and that we appreciate all the suggestions of Reviewer 1.

Reviewer 2 Report
This article is devoted to the extracts of Rhodiola rosea L. and their biologically active properties. This article is new and current. Studies on the biological activity of some plant extracts have intensified some time ago against the backdrop of the COVID pandemic. In terms of subject matter, volume and quality of data, this article corresponds to the subject of the journal. There are some points for improvement:
1. It is necessary to unify the drawings.
2. It is necessary to clarify where the extract of Rhodiola rosea was taken from. If the authors use the purchased version, indicate the manufacturer. If the authors independently obtain an extract, it is necessary to indicate the method of obtaining.
3. It is desirable to analyze the extract to determine the main components. All of them are presented in different quantities and have different degrees of biological activity.
4. Please cite: 10.3390/molecules27186129.
5. It is advisable to write short conclusions on this work.
Author Response
Reviewer 2:
- It is necessary to unify the drawings.
Authors´ responses:
All the graphs in the manuscript were made by one author and same colours, legend and description was made uniformly. However we changed Figure 1 in ax-x to be exactly in the same format as others Figures 2-5. We hope that this adjustment fits your expectation.
----------------------------------------------------------------------------------------
Reviewer 2:
- It is necessary to clarify where the extract of Rhodiola rosea was taken from. If the authors use the purchased version, indicate the manufacturer. If the authors independently obtain an extract, it is necessary to indicate the method of obtaining.
Authors´ responses:
Thank you for this question, The authors would like to clarify, (as this is specified in section “4.5 Phytochemical composition of the Rhodiola rosea L. dry extract (RSE)” that this “Rhodiola rosea L. dry extract was manufactured by Calendula, Slovakia, and gifted as a courtesy to the Faculty of Pharmacy at Comenius University in Bratislava.”Here we see the information gap, and we decided to add a sentence, which bridges our experiment with the manufactured extract: “Flowingly, the Rhodiola rosea L dry extract, batch No. 3-049-001-06-18, was than gifted to our department, where one part of this batch was tested at Faculdade de Farmácia da Universidade de Lisboa, Lisbon, Portugal for testing on the collagen-induced arthritis and the second part of the same batch was tested at the Centre of Experimental Medicine SAS, Institute of Experimental Pharmacology & Toxicology, v.v.i, Bratislava, Slovakia for experiments on the adjuvant-induced arthritis.”
Moreover coworkers from Faculty of Pharmacy at Comenius University in Bratislava were project members of common project APVV-15-0308 and the portugal coworkers were also project members of bilateral project APVV-SK-PT-18-0022. These collaborations allows us to share and study the extract Rhodiola rosea L. manufactured by Calendula.
-----------------------------------------------------------------------------------------
Reviewer 2:
- It is desirable to analyze the extract to determine the main components. All of them are presented in different quantities and have different degrees of biological activity.
Authors´ responses:
Thank you for this question. The following specification for Rhodiola rosea L., herbal preparation is found for the monograph: Dry extract (DER 1.5-5:1), extraction solvent ethanol 67-70% v/v, with a footnote pointing to the requirement that a narrow range of the DER and a fixed strength for the ethanol used for extraction need to be specified for each herbal medicinal product. Because the batch No. 3-049-001-06-18 was not a finished commercial medicinal product but the powder of Rhodiola rosea L. was an intermediate product, the analytical requirements followed the standard quantitative and qualitative composition of Rhodiola rosea L. This is also with regard to the registration application of Article 16d(1) of Directive 2001/83/EC as amended and the Community herbal monograph on Rhodiola rosea L. This is, however, required for Rhodiola rosea L. medicinal product intended for indication as an adaptogen. Because the Community herbal monograph on Rhodiola rosea L. do not require other further analysis, the analytical certificate from Calendula manufacturer do not consequently include no other specifications to determine other relevant components. This approach has been agreed across the EU, if all manufacturing processes remain exactly the same to maintain the approved adaptogen indication. The important aspect is, that the content of salidroside indicates the quality and quantity for the adaptogen indication. This requires the Community herbal monograph on Rhodiola rosea L. and the analytical certificate has followed this requirement as well. In our case, the authors decided to add the original analytical certificate by Calendula as supplementary material to the manuscript for transparency. Moreover, the authors are also curious, which component was responsible for the antiarthritic effect, and specifically for the Rhodiola rosea L., dry extract (DER 1.5-5:1), with extraction solvent ethanol 67-70% v/v. Therefore we have decided to request our colleagues at the Faculty of Pharmacy at Comenius University in Bratislava to perform further detailed analysis, so we may study in the future also other components like polyphenols. This, however, need to be covered by a new scientific grant and preparing a new preclinical experiment to study the relationship between the analysed extract and its pharmacologic activity. Polyphenols study will be our new contribution. In the experimental design will include determination of polyphenols, salidroside and original extract for comparing their antiarthritic effect.
-----------------------------------------------------------------------------------------
Reviewer 2:
- Please cite: 10.3390/molecules27186129.
Authors´ responses:
The Authors have add this citation (Demirpolat, A.; Akman, F.; Kazachenko, A.S. An Experimental and Theoretical Study on Essential Oil of Aethionema sancakense: Characterization, Molecular Properties and RDG Analysis. Molecules 2022, 27, 6129. https://doi.org/10.3390/molecules27186129) into the text of conclusion and references.
We intend to use this publish method for the next analytical approach as we have mentioned above.
-----------------------------------------------------------------------------------------
Reviewer 2:
- It is advisable to write short conclusions on this work.
Authors´ responses:
The authors would like to thank for this advice and we decided to separate the conclusion part from the discussion to have the text more readable for others.
-----------------------------------------------------------------------------------------
We hope all issues were addressed and met for this manuscript. We believe that the manuscript has now been sufficiently improved, and therefore, the authors would also like to express their thanks and that we appreciate all the suggestions of Reviewer 2.
